# Eosinophils in the Gastrointestinal Tract: Key Contributors to Neuro-Immune Crosstalk and Potential Implications in Disorders of Brain-Gut Interaction

**DOI:** 10.3390/cells11101644

**Published:** 2022-05-14

**Authors:** Eloísa Salvo-Romero, Bruno K. Rodiño-Janeiro, Mercé Albert-Bayo, Beatriz Lobo, Javier Santos, Ricard Farré, Cristina Martinez, María Vicario

**Affiliations:** 1Translational Mucosal Immunology Group, Digestive System Research Unit, Vall d’Hebron Institut de Recerca (VHIR), Vall d’Hebron Hospital Universitari, Passeig Vall d’Hebron 119-129, 08035 Barcelona, Spain; eloisasalvo1@gmail.com (E.S.-R.); merce.albert@vhir.org (M.A.-B.); 2Laboratory of Neuro-Immuno-Gastroenterology, Digestive System Research Unit, Vall d’Hebron Institut de Recerca (VHIR), Vall d’Hebron Hospital Universitari, Passeig Vall d’Hebron 119-129, 08035 Barcelona, Spain; brunokotska@hotmail.com (B.K.R.-J.); beatriz.lobo@vhir.org (B.L.); javier.santos@vhir.org (J.S.); 3Facultad de Medicina, Universitat Autònoma de Barcelona, 08193 Bellaterra, Spain; 4Department of Gastroenterology, Vall d’Hebron Hospital Universitari, Passeig Vall d’Hebron 119-129, 08035 Barcelona, Spain; 5Centro de Investigación Biomédica en Red de Enfermedades Hepáticas y Digestivas (CIBEREHD), 28029 Madrid, Spain; ricard.farre@kuleuven.be; 6Department of Chronic Diseases and Metabolism (CHROMETA), Translational Research Center for Gastrointestinal Disorders (TARGID), KU Leuven, 3000 Leuven, Belgium; 7Vascular and Renal Translational Research Group, Lleida Institute for Biomedical Research Dr. Pifarré Foundation (IRBLleida), Av. Alcalde Rovira Roure 80, 25198 Lleida, Spain; 8Department of Gastrointestinal Health, Nestlé Institute of Health Sciences, Société des Produits Nestlé S.A., Nestlé Research, Vers-chez-les-Blanc, 1000 Lausanne, Switzerland

**Keywords:** intestinal eosinophils, neuro-immune interaction, disorders of brain–gut interaction

## Abstract

Eosinophils are innate immune granulocytes actively involved in defensive responses and in local and systemic inflammatory processes. Beyond these effector roles, eosinophils are fundamental to maintaining homeostasis in the tissues they reside. Gastrointestinal eosinophils modulate barrier function and mucosal immunity and promote tissue development through their direct communication with almost every cellular component. This is possible thanks to the variety of receptors they express and the bioactive molecules they store and release, including cytotoxic proteins, cytokines, growth factors, and neuropeptides and neurotrophines. A growing body of evidence points to the eosinophil as a key neuro-immune player in the regulation of gastrointestinal function, with potential implications in pathophysiological processes. Eosinophil–neuron interactions are facilitated by chemotaxis and adhesion molecules, and the mediators released may have excitatory or inhibitory effects on each cell type, with physiological consequences dependent on the type of innervation involved. Of special interest are the disorders of the brain–gut interaction (DBGIs), mainly functional dyspepsia (FD) and irritable bowel syndrome (IBS), in which mucosal eosinophilia and eosinophil activation have been identified. In this review, we summarize the main roles of gastrointestinal eosinophils in supporting gut homeostasis and the evidence available on eosinophil–neuron interactions to bring new insights that support the fundamental role of this neuro-immune crosstalk in maintaining gut health and contributing to the pathophysiology of DBGIs.

## 1. Introduction

Eosinophils are innate immune granulocytes involved in the defense against parasites and in the pathogenesis of Th2 immune-mediated disorders [1]. During the last decade, the prevailing paradigm indicating that eosinophils act as strictly pro-inflammatory end-stage effector cells is being progressively replaced by cumulative observations showing that eosinophils are multifaceted fine-tuned tissue residents that are involved in multiple phases of the immune response, participate in tissue regeneration, and play a role in homeostasis maintenance [2,3,4]. The wide variety of receptors expressed by eosinophils and their ability to produce and release a myriad of mediators enable them to develop tight and complex bi-directional communication with almost every tissular component, including epithelial immune cells and neurons (Figure 1), as recently described in the context of asthma and allergic respiratory inflammation [3,5]. Consequently, resident and newly recruited eosinophils may respond to specific tissue requirements. This response implicates the regulation of immunoglobulin production, the promotion of both anti- and pro-inflammatory networks, tissue remodeling and repair, and modulation of epithelial barrier function at mucosal sites [6,7,8]. Notably, the gastrointestinal tract harbors the largest number of resident eosinophils in the body and eosinophilia and eosinophil activation are associated with gut dysfunction in a variety of inflammatory diseases, such as eosinophilic gastrointestinal diseases (EGIDs) and inflammatory bowel disease (IBD), and in other non-inflammatory entities, such as the disorders of brain–gut interaction (DBGIs). Eosinophil plasticity and the capacity to communicate with the nervous system support additional pathways by which eosinophils contribute to gut homeostasis but also to disease mechanisms and symptom generation [9,10] which present potential avenues for therapeutic intervention.

In this review, we provide a comprehensive description of the role of eosinophils in the gastrointestinal tract, focusing on neuro-immune mechanisms that associate eosinophil activity with gut homeostasis and disease. We also summarize current knowledge on eosinophil–nerve communication in common DBGIs and support the need for further investigation of eosinophils as potential therapeutic targets to promote gut health.

## 2. Eosinophils in the Gastrointestinal Tract

Eosinophils are continuously produced in the bone marrow from pluripotent CD34+ stem cells. Under the control of specific transcription factors (such as Δdbl-GATA-1, PU.1, and C/EBP) and in response to IL-5 [11], mature eosinophils are released into the peripheral circulation and then transferred to target tissues, being the gastrointestinal tract their main destination with the only exception of the healthy esophagus where the presence of eosinophils is a histological hallmark of pathology such as in eosinophilic esophagitis (EoE) [12].

The gastrointestinal tract is constantly exposed to multifactorial pressures (both internal and environmental) that are ultimately responsible for shaping and diversifying intestinal eosinophil identity, presumably with differential impacts on gut function and disease outcomes. Differences in eosinophil morphology, surface phenotypes, and localization within the gastrointestinal tract have been described both at the macro-level (small vs. large bowel) as well as at the level of tissue microdomains (crypts vs. villi, intraepithelial vs. *lamina propria*). Human data regarding eosinophil sub-populations in the gastrointestinal tract are scarce, however, and most of the available information regarding intestinal eosinophil phenotypic diversity has been gained from the study of mouse models (exhaustively reviewed in [13]).

Intestinal eosinophils can be distinguished from the bone marrow and circulation pools by the surface receptors they express. Steady-state intestinal eosinophils constitutively express elevated levels of CD11c, Siglec F (Siglec 8 in humans), and CD11b, resembling inflammatory eosinophils recruited to the airways in the context of allergic asthma in both humans and mouse models [14,15]. Recruitment of eosinophils to the intestinal mucosa is critically dependent on local production of chemoattractants, the most selective one being the chemokine eotaxin-1 (C-C motif ligand 11, CCL11) constitutively expressed along the gastrointestinal tract [12], followed by eotaxin-2 (CCL24), eotaxin-3 (CCL26), RANTES (CCL5), and monocyte chemoattractant protein-3 (MCP-3), which bind to their C-C chemokine receptors (CCRs) (CCR1, CCR3, and CCR4) [16]. In addition, adhesion molecules are also involved in intestinal eosinophil recruitment towards the gastrointestinal tract. While eosinophils expressing the surface receptor integrin α4β7 are preferentially directed towards the small intestine by mucosal vascular addressin cell adhesion molecule 1 (MadCAM1) [17], the recruitment of eosinophils to the colon seems to be more dependent on intercellular adhesion molecule 1 (ICAM1) expression [18].

Although there is evidence showing that eosinophils specifically recruited to the small bowel exhibit particular phenotypic characteristics, such as uniquely prolonged longevity [19] and compartmentalized morphological heterogeneity [20,21], the functional significance of these sub-populations of differentially recruited eosinophils remains elusive. One of the most characteristic features of eosinophils is their possession of cytoplasmic granules containing stored mediators which equip them to quickly respond to the local requirements of the tissues in which they reside, enabling them modulate tissue adaptability to environmental changes. Eosinophil granules contain a considerable variety of preformed biologically active substances (cytotoxic cationic proteins, cytokines, growth factors, chemokines, neuropeptides, and enzymes) ready for rapid release, and eosinophils are also able to synthesize *de novo* different mediators upon eosinophil stimulation (Table 1). However, it is unknown whether distinct eosinophil sub-phenotypes harboring differential profiles of surface receptors and granule contents may determine unique specific functions within particular gut tissue regions or compartments, as has been described for mast cells (reviewed in [22]).

### 2.1. Eosinophils Regulate the Intestinal Barrier Function

Although the impact of eosinophils on the intestinal epithelium in *in vivo* human studies is difficult to decipher, data gained from *in vitro* and pre-clinical models have shown a significant influence of eosinophils on mucosal homeostasis (Figure 2).

The intestinal epithelium exerts a broad variety of functions ranging from simply being a physical barrier to luminal content to directly promoting active defensive immunity, achieved thanks to the existence of bi-directional communication between intestinal epithelial cells and the whole plethora of mucosal resident immune cells. Eosinophil-induced promotion of intestinal barrier protection seems to depend on the abundance of eosinophils, a low eosinophil–epithelial cell ratio being associated with enhanced barrier integrity [26,27,28], while a total depletion of intestinal eosinophils is associated with barrier disruption, as experimentally demonstrated by a high-fat diet treatment [29]. A remarkable exception to this pattern is the case of the healthy esophagus, where increased eosinophil numbers in EoE are linked to chronic inflammation and barrier disruption [30]. Notably, the close proximity of eosinophils and intestinal epithelial cells facilitates the crosstalk between these two cell types (Figure 1) which may evoke a wide range of effects, ultimately contributing to maintenance of homeostasis or barrier dysfunction.

The interplay between eosinophils and the airway epithelium has been extensively studied, and similar interactions may be expected in the intestinal mucosa. Indeed, a large number of studies have demonstrated that, similar to the airways, intestinal epithelial cells are potent sources of immune-modulating cytokines, chemokines, and growth factors, with large effects on eosinophil recruitment, survival, and activation [12,19]. On the other hand, eosinophils can signal to epithelial cells through the generation of reactive oxygen species (ROS) and the release of leukotrienes and other granule-stored mediators, thus augmenting local inflammation and promoting epithelial proliferation [31,32]. In EoE, eosinophil-derived mediators have been shown to target esophageal epithelial integrity. More specifically, IL-13 down-regulates the intercellular adhesion molecule desmoglein-1, but not desmoglein-3, leading to altered epithelial barrier function and the promotion of inflammatory responses [33,34]. Eosinophils also modulate the secretion of mucus and bactericidal substances contributing to the reinforcement of epithelial barrier protection against infection. In fact, eosinophil-deficient mice show significantly reduced numbers of mucus-secreting goblet cells in the small bowel [35]. Moreover, supernatants derived from activated eosinophils increase mucin levels in human airway epithelial cells [36,37], and blocking the eotaxin receptor with anti-CCR3 monoclonal antibody in a mouse model of asthma reduces lung eosinophil recruitment and mucus overproduction [38]. Additionally, eosinophils have the ability to synthesize and rapidly release a web-like complex meshwork of DNA fibers and granule proteins called eosinophil extracellular traps (EETs) in a ROS-dependent manner but independently of eosinophil death, creating a second physical barrier that limits bacterial invasion [39]. However, EETs could also have deleterious effects on intestinal health due to increased production of mucus secretions [40]. Indeed, EETs have been associated with respiratory diseases and a number of other inflammation-associated diseases affecting the skin (atopic dermatitis, urticaria) and the gastrointestinal tract (IBD and EoE) [40].

Additional to the direct effect on the epithelium, eosinophils also contribute to barrier maintenance by promoting tissue remodeling/repair after tissue damage induced by pathogens, toxins, or cell death. A dual role of eosinophils in tissue degradation and repair has been described. In fact, eosinophils release eosinophil-derived neurotoxin (EDN), which promotes fibroblast proliferation [41], and major basic protein (MBP), which synergizes with IL-5 and transforming growth factor beta (TGF-β) to enhance fibroblast expression and secretion of IL-6 and IL-11 [42]. In addition, MBP and EDN are potent heparanase inhibitors, which suggests that eosinophils may be involved in proteoglycan degradation prevention and contribute, therefore, to proteoglycan accumulation in fibrotic tissues [43]. Tissue degradation can be sensed by eosinophils which react to promote healing by facilitating epithelial proliferation and acting on the vasculature through the release of vascular endothelial growth factor, fibroblast growth factor, and transforming growth factor-β1 (TGF-β1) [2] in order to recover tissue structure and promote homeostasis.

### 2.2. Eosinophils Modulate Intestinal Immune Responses

Eosinophils are a major cellular element of the intestinal mucosa and participate in the regulation of immunity and in the promotion of inflammatory responses (Figure 2). They are omnipresent at sites of Th2-mediated inflammation, where they can precede or be rapidly recruited to tissue, independently of adaptive immune responses [44]. This suggests that eosinophils themselves can modulate and/or sustain the Th2 character of the local tissue immune microenvironment and also promote Th2 memory cell functions by, for example, releasing cytokines [45]. Beyond Th2 immunity, preformed eosinophil mediators are also related to other immune responses; these include cytokines associated with Th1 (IFN-γ, IL-12), T-regulatory (IL-10, TGF-β), and Th17 [46,47] activities. In addition, eosinophils have the capacity to initiate and polarize adaptive immune responses. Unlike their blood counterparts, intestinal eosinophils constitutively express on their cell surface antigen presentation markers, including MHC class II, CD80, and activating receptor FcγRIII, suggesting that gut eosinophils may be primed for antigen presentation [20,48,49]. In addition, eosinophils also promote the initiation of adaptive immune responses through the granule proteins EDN and EPO which are involved in dendritic cell migration, activation, and maturation, leading, therefore, to enhanced antigen-specific Th2 responses [50]. Notably, eosinophils contribute to immunoglobulin production by plasma cells, as shown in animal studies by mediating adjuvant-elicited priming of B cells and optimal antigen-specific early IgM through IL-4 [51]. Eosinophils are crucial for class-switch generation and the maintenance of IgA plasma cells in the *lamina propria*, mainly through the expression of the A proliferation-inducing ligand (APRIL) and other cytokines, as identified in models of eosinophil-depleted mice [6].

Eosinophil-derived cytokines and chemokines also have effects on innate immune cells, especially on mast cells. Eosinophil–mast cell interactions have been the focus of many hypotheses trying to explain mucosal immune responses (reviewed in [52]). The large amount of evidence of eosinophil and mast cell contribution in the same scenarios, along with the important role mast cells play in allergic and inflammatory diseases also related to the brain–gut axis, suggest an additional eosinophil-derived effector function in immunoregulation within this axis. This potential contribution should no longer be interpreted as a minor interaction based on the release of eosinophil proteins and their inflammatory activity. In fact, eosinophils can promote mast cell growth, survival, and activation by several cytokines and by granule-derived proteins, such as MBP, stem cell factor (SCF), and nerve growth factor (NGF) [52,53]. Additionally, mast cells secrete mediators needed for reciprocally activating and promoting the survival of eosinophils [53]. However, mast cells may not be necessarily required for eosinophil survival, as in a mast cell-deficient mouse model of EoE the number of eosinophils remain unaffected [54].

### 2.3. Eosinophil–Neuron Interactions

#### 2.3.1. Innervation of the Gastrointestinal Tract

The gastrointestinal tract is densely populated by two complex networks of neurons (intrinsic innervation) and immune cells that have co-evolved mechanisms to sense and rapidly adapt to the highly dynamic environmental challenges taking place at the intestinal mucosa. The first network is the submucosal plexus or Meissner’s plexus, located in the submucosal region between the circular smooth muscle and the submucosa, which controls glandular secretions, regulates local blood flow, and controls water secretion into the lumen. Fundamental research in different species, such as guinea pigs, rats, and humans, has shown that secretomotor neurons release acetylcholine (ACh) and vasoactive intestinal polypeptide (VIP) which stimulate chloride (Cl-) and water secretion. Nevertheless, the findings in mice are contradictory. In this animal species, the activation of submucosal neurons by electrical field stimulation involves in part the release of Ach and the activation of muscarinic receptors; however, the pharmacological stimulation of these neurons with veratridine does not involve the release of Ach [55]. The second network, called the myenteric plexus or Auerbach plexus, is located between the circular and the longitudinal smooth muscle layers and plays a crucial role in controlling gastrointestinal motility. Excitatory motor neurons synthetize and release ACh as the main neurotransmitter, along with substance P (SP) and other tachykinins. Inhibitory motor neurons synthetize nitric oxide, the main inhibitory neurotransmitter in the upper gastrointestinal tract, and adenosine triphosphate (ATP), the main inhibitory neurotransmitter in the lower gastrointestinal tract. Both plexi of the enteric nervous system (ENS) operate independently but are in turn modulated by the autonomic nervous system via efferent sympathetic and parasympathetic innervation (extrinsic innervation). The cell bodies of these nerves are located in the celiac, superior, and inferior mesenteric ganglions of the sympathetic chain and in the brain, respectively. The central nervous system (CNS) processes sensory information from the different layers of the gastrointestinal tract via the vagal afferent (non-painful physiological stimuli) and spinal afferent nerves (nociceptive stimuli—pain). Gastrointestinal symptoms are triggered basically through the stimulation of chemosensitive nociceptors present in spinal afferent nerves that innervate the *lamina propria* and through the activation of mechanosensitive nociceptors present in the longitudinal and circular smooth muscle [56]. The activation of spinal afferent nerves induces the release of sensory neuropeptides, such as SP, CGRP, and NKA.

How the different branches of the extrinsic and intrinsic nervous system crosstalk with innate and adaptive immune cells residing in the gut to jointly coordinate critical physiological functions and responses to challenges has been a matter of intense research. Indeed, functional neuro-immune interactions have been described as playing fundamental roles in intestinal health and disease [57] and there are excellent reviews discussing neuronal crosstalk in several gut immune populations (mainly mast cells, macrophages, and T- and B-cells) [9,10]. However, the role of eosinophils in this intestinal neuro-immune axis has systematically received much less attention and most of the knowledge we currently have regarding the influence of eosinophil activity on the nervous system has been gained in the airways in the context of asthma and allergic respiratory inflammation [3,5]. It is clear from an anatomical/functional point of view that the (patho)physiological consequences of eosinophil–neuron communication will depend on the type of innervation (intrinsic or extrinsic) involved.

Like other gut resident immune cells, eosinophils express a wide range of neuropeptides and their receptors (Table 2) that confer on them abilities of interacting with the nervous system, either directly by cell-to-cell contact or indirectly through eosinophil crosstalk with other immune cells residing in the *lamina propria* [47,57]. This eosinophil–neuron communication has been shown to have a bi-directional nature (Figure 3); eosinophils can be either the source or the target of the interactions. Indeed, neurons can recruit and activate eosinophils, while eosinophils have been described to show trophic, stimulatory, and inhibitory effects on neurons.

#### 2.3.2. Neural-Induced Recruitment and Activation of Eosinophils by Extrinsic Nerves

Different *in vitro* and *in vivo* studies have shown that airway nerves actively recruit and bind eosinophils, promoting a wide range of effects on both cell types. Direct recruitment of eosinophils can happen through eotaxins constitutively expressed and released by parasympathetic efferent nerves, as occurs in the airways during antigenic challenge [58,59]. In addition, neurotransmitters and sensory neuropeptides released from the peripheral nerve endings of sensory neurons (extrinsic innervation), such as SP, CCK-8, NKA, and CGRP, can also promote eosinophil recruitment [60,61,62,63]. After recruitment, eosinophils adhere to nerves through cell adhesion molecules (CAMs), which leads to eosinophil activation and degranulation [61,63].

**Table 2 cells-11-01644-t002:** Neuropeptide receptor expression and effects on eosinophils.

Receptor ^1^	Effect	References
Adenosine receptors A1, A2a, and A3	Activation (A1), NADPH oxidase activity regulation (A2a), pro- or anti-inflammatory response (A3)	[64,65]
Adrenergic receptors (α1, α2), β1, β2, and β3	Inhibition of NADPH oxidase, degranulation, chemotaxis, adhesion molecules, membrane lipid metabolism (β2). Inhibition of ICAM adhesion, ROS production, and EDN degranulation in IL-5, LTD4, and CXCL10 primed eosinophils	[64,66,67,68]
Bradykinin receptor B1, B2	Proliferation, migration, and increase in intracellular Ca^2+^ levels, generation of lipid bodies and decreased eosinophil cell count in allergic airway inflammation (B1) Implications for eosinophil accumulation (B2)	[64,69,70,71]
Cannabinoid receptor CB2	Chemotaxis, ICAM adhesion, increased eotaxin-2-primed CD11b expression, increased ROS production	[72,73,74]
Calcitonin gene-related peptide (CGRP) receptor	Increased migration	[60,64]
Histamine receptors H1R, H2R, H4R	Inhibition of ROS, EPO release, and chemotaxis (H_2_R) Priming of chemotaxis to eotaxin and adhesion to endothelium (H_4_R)	[75,76,77,78]
Muscarinic receptors M2, M3	Stimulate production and release of CRF	[79]
Nicotinic acetylcholine receptors (nAChRs) α-3, -α4, and α-7	Decrease infiltration into the lungs and airways Down-regulate eosinophil function *in vitro*	[80]
Purinergic receptors 2 P2Y and P2X family	Chemotaxis, induction of ROS production, CD11b upregulation, calcium mobilization, production of cytokines and ECP, induce release of EDN, EPO, and inflammatory factors	[81,82,83,84]
Serotonin receptor 5-HT1 (A, B, E), 5-HT2A	Migration(5-HT2A); effects on rolling and changes in shape of eosinophils	[85,86]
Tachykinin receptor NK1, NK2, and NK3	Induction of the expression (NK1) and secretion (NK2) of CRF Increment of ROS production and thromboxane and degranulation of eosinophils	[64,87,88,89]
Vasointestinal peptide associated receptor CRTH2	Chemokinesis or chemotaxis	[64,90,91]

^1^ CD11b, Cluster of differentiation molecule 11b; CRF, Corticotropin releasing factor; CRTH2, Chemoattractant receptor-homologous molecule expressed on Th2 cells; CXCL10, C-X-C motif chemokine 10; EDN, Eosinophil derived neurotoxin; EPO, Eosinophil peroxidase; ICAM, Intercellular adhesion molecule; IL, Interleukin; LTD4, Leukotriene D4; PAF, Platelet-activating factor; ROS, Reactive oxygen species.

After recruitment, eosinophils adhere to nerves through cell adhesion molecules (CAMs), which leads to eosinophil activation and degranulation [61,63]. Two major CAM families are involved in eosinophil–nerve binding: the immunoglobulin superfamily of CAMs, which include vascular, intercellular, and neural cell adhesion molecules (VCAM, ICAM, and NCAM); and integrin CAMs, mainly integrin α4β1/VLA-4 and leukocyte function-associated molecule 1 (LFA1). CAM expression on nerves can be constitutive or inducible by pro-inflammatory cytokines and other mediators. Indeed, ICAM expression is induced by TNF-α and IFN-γ causing increased eosinophil adhesion and switching binding preference from VCAM to ICAM in primary cultures of airway parasympathetic nerves [92], an effect that is prevented by treatment with dexamethasone and NFKB inhibitors [93]. Eosinophil engagement to neural adhesion molecules leads to eosinophil activation and ROS production [94,95]. Indeed, ICAM and VCAM activation concurrently induce neurite retraction via the generation of tyrosine kinase-dependent ROS and by the p38 MAP kinase pathway [96]. At the same time, neural-derived ROS trigger eosinophil degranulation [56], which makes ROS production a shared event in the eosinophil–nerve bi-directional interaction.

Additional neural stimuli, such as ACh released by peripheral efferent nerves, have been linked to eosinophil chemotaxis and degranulation in experimental models [97] and in atopic asthma patients [98]. It is likely that a similar mechanism exists in the gastrointestinal tract, but further investigations are needed to define the eosinophil–nerve interactions and their contributions to organ functions.

#### 2.3.3. Neural-Induced Recruitment and Activation of Eosinophils by the ENS

Although the anatomy and function of extrinsic nerves are very different, eosinophil–enteric nerve interactions seem to be similar to those described in the parasympathetic efferent innervation of the lungs. Nevertheless, we currently have little evidence about the role of the ENS in recruiting and activating eosinophils. ICAM and eotaxin-3 are overexpressed in ganglia of the myenteric plexus in refractory IBD patients. Interestingly, eosinophils are in close proximity to terminal varicosities of excitatory motor neurons expressing SP and choline acetyltransferase (ChAT) but not neuronal nitric oxide synthase (nNOS) [87]. In addition, another cell adhesion molecule, NCAM, has been reported to play a role in eosinophil adhesion to myenteric terminal varicosities in the colonic mucosa of rats undergoing a Th2 response caused by parasitic infection [99]. However, the specific mechanisms underlying eosinophil adhesion to excitatory terminal varicosities during intestinal inflammation are still unknown.

In the gastrointestinal mucosa, SP signaling has been shown to trigger the production and release of corticotropin releasing factor (CRF) *in vitro* and in experimental chronic restraint stress models [79,100]. CRF is a major mediator of stress-induced autonomic, hormonal, and behavioral reflexes that inhibit inflammatory responses at regional levels and influence gut motility and secretion [101] and it has been recently involved in a non-classical non-pro-inflammatory eosinophil activation mechanism, wherein neuromediators selectively induce eosinophil synthesis and release of CRF by piecemeal degranulation [102]. VIP is a key signaling molecule in the neuro-immune network that is secreted by neuronal cells and by different types of immune cells and exerts a wide spectrum of functions. The release of VIP by enteric neurons innervating the intestinal mucosa modulates the epithelial barrier [103]. VIP also regulates the production of both anti- and pro-inflammatory mediators in immune cells [91]. Eosinophils do not express the classical VPAC-1 and VPAC2 but respond to VIP through the Chemoattractant Receptor-Homologous Molecule Expressed on Th2 Cells (CRTH2) [90]. The main effects of VIP on eosinophils are the promotion of chemotaxis and the production of prostaglandin D2, both identified in the context of allergic rhinitis. It may be relevant to determine whether eosinophilia is promoted by VIP released by enteric neurons during intestinal inflammatory conditions, and, additionally, whether eosinophils also impact the intestinal barrier by means of VIP, as they can store and release this neuropeptide [104].

#### 2.3.4. Eosinophil-Dependent Neuroplasticity

Among the repertoire of molecules expressed by eosinophils, there are multiple neuromodulatory mediators, including neurotrophins and neuropeptides (Table 1), that confer on eosinophils the ability to influence nerve growth, reactivity, survival, and neurotransmitter release. Indeed, eosinophil–neural crosstalk induces dual effects on diverse neural plasticity mechanisms that lead to aberrant neurotransmission, including both increased neural growth and activation and, contrarily, growth inhibition and nerve damage. On the one hand, increased neural activity appears to be associated with the cardinal symptoms of eosinophil-mediated chronic inflammatory respiratory and dermatological diseases, such as pain, cough, itch, and bronchoconstriction [105,106]. Conversely, eosinophil degranulation may also mediate loss of neural activity and neuropathic damage in demyelinating diseases [107].

Eosinophil influence on neural growth and damage has been also observed in the intestinal mucosa in experimental models and in human tissue. Increased eosinophil recruitment to enteric nerves, together with higher densities of nerves expressing the growth and plasticity marker growth-associated protein 43 (GAP43), have been described in an enteric parasitic infection pre-clinical model [99]. Moreover, eosinophils locate next to vacuolated and necrotic axons in the jejunal mucosa in experimental enteric eosinophilic inflammation [108], suggesting a neuropathic interaction with myenteric nerves. Similarly, studies on acute appendicitis have reported elevated numbers of eosinophils and mast cells, along with significant increases in nerves and ganglion cells, providing a possible mechanism underlying pain generation in these patients [109].

Eosinophils also promote cholinergic neuronal remodeling, a feature in many inflammatory diseases. In human and animal models of asthma, eosinophils are actively recruited and activated by nerves via tachykinins causing MBP release and a consequent dysregulation of the vagal muscarinic M2 receptor [110], which contributes to ACh release, nerve hyperactivity, and/or nerve remodeling, altering vagal-mediated smooth muscle contraction responses [61]. *in vitro* data have shown that eosinophil adhesion to IMR32 neural cells promotes the release of eosinophil granule proteins that confer nerve plasticity modulation capacities to protect nerves from inflammation-associated injury, mainly through MBP-dependent abolition of apoptosis [92,95,111,112]. This crosstalk between eosinophils and parasympathetic nerves described in asthma has also been highlighted as a potential contributor to intestinal pathology. In human intestinal biopsies, mucosal eosinophils and eosinophil granule proteins co-localize with nerves and myenteric ganglia [113,114]. Moreover, in the colonic mucosa, eosinophils express M2 and M3 receptors and release CRF upon cholinergic activation, which disrupts intestinal epithelial barrier function through a paracrine effect on mast cells [79].

## 3. Emerging Role of Eosinophils in Disorders of Brain–Gut Interaction

A recent large-scale multinational study has shown that DBGIs affect around 40% of the population worldwide, that they are more frequent in women than men, and that they are associated with a poor quality of life [115], leading to major economic impacts on health systems. Despite intense research during the last 20 years, these disorders are still diagnosed only by clinical criteria and by exclusion of organic diseases that may produce similar symptoms. Among these disorders, functional dyspepsia (FD) and irritable bowel syndrome (IBS) are among the most prevalent. These entities are characterized by chronic complaints arising from altered brain–gut interactions leading to intestinal dysmotility and visceral hypersensitivity [116]. The pathophysiology of these disorders is complex and not yet well understood; however, both FD and IBS show epithelial barrier dysfunction and mucosal immune activation which, in a subset of patients, are associated with major clinical complaints and their severity [117,118]. The trigger for the subtle immune activation is unknown, but presumably food, acid, bile salts, and microbiota are involved [119,120]. Indeed, observational studies focused on pathophysiological research report increased epithelial permeability and low-grade duodenal mucosal immune eosinophil and/or mast infiltration in up to 40% of patients with FD [121]. However, eosinophil or mast cell number and activation (as measured by granular complexity and degree of degranulation) seem not to correlate with duodenal epithelial integrity (as measured by transepithelial electrical resistance (TEER)) in these patients [122]. In fact, an altered barrier function but, contrary to previous studies, not increased eosinophil or mast cell infiltration, has recently been confirmed in two small cohorts of patients with FD, using standard histological methods. One study, using confocal laser endomicroscopy showed significantly higher epithelial gap density, impaired permeability to ions assessed by TEER measurements, and a quantitative increase in inflammatory cell death (pyroptosis), along with significantly higher gene expression of IL-6 in the duodenal mucosa [123]. Notably, permeability correlated with the severity of certain dyspeptic symptoms. A second study showed a modest reduction in the expression of several duodenal tight junction and adherens junction proteins, which may be secondary to up-regulation of regulatory miRNAs and the increase in small intestinal permeability measured *in vivo* [124]. A recent study, measuring impedance baseline by using a non-validated custom-built catheter-based technique, confirmed impaired duodenal and jejunal mucosal barrier function in FD [125].

Unfortunately, there are still only limited data available by means of which to define how eosinophils participate in these networks to modulate gut function. Some reports show a context of enteric nerve dysfunction associated with eosinophil accumulation in mucosal samples from both FD and IBS patients. In FD patients, a neuronal functional impairment of the submucosal plexus of duodenal biopsies has been reported, with increased expression of glial markers and altered ganglionic architecture and signaling, together with an increased infiltration of eosinophils and mast cells in the submucosal plexus [113]. Interestingly, the authors found a negative association between the number of eosinophils and neuronal function. Additionally, increased mucosal infiltration and degranulation of duodenal eosinophils have been shown to promote enteric nerve fiber density and sprouting in patients with FD [114], a finding that supports a key role of this granulocyte through neuro-immune mechanisms in this disorder.

In IBS, research has mainly shown the involvement of mast cell–nerve interaction in hyperalgesia [22,126] and the association of mast cell activation with epithelial barrier dysfunction [127], but a role for eosinophils in neuro-immune networks and eosinophil impact on epithelial permeability has not yet been defined. Mucosal eosinophil counts do not show agreement in different intestinal segments and in particular subsets of IBS patients [128]. Rather than cell number, the activation of eosinophils may offer a new perspective for exploring their potential contribution to IBS pathophysiology. Recent work in our group showed a higher degree of piecemeal degranulation and higher granule CRF contents correlated with cardinal clinical and psychological manifestations of diarrhea-predominant IBS (IBS-D), despite eosinophil infiltrations similar to those of healthy controls [102]. Moreover, it has been shown in mouse and in *in vitro* models that eosinophils respond to SP and carbachol (a cholinergic agonist) by increasing secretory activity and CRF synthesis and release [79,100] without promoting pro-inflammatory activity—a profile similar to that found in mucosal eosinophils from IBS-D [102]. Thus, the release of SP or ACh from free nerve endings located in the *lamina propria* may potentially activate gut eosinophils to produce and release CRF, mediating local neuro-immune circuits in stress-related intestinal pathology. These results highlight the need for further research to define the potential contribution of eosinophils to gastrointestinal dysfunction, though it is difficult to separate the effects related only to the IBS physiopathology or psychological comorbidity.

Although there is a wide range of therapeutics for treating eosinophilic disorders [129,130], including oral and systemic corticosteroids, monoclonal antibodies, and immunosuppressants, the recent description and involvement of these cells in DBGIs has not yet allowed researchers to define their potential advantages in the management of these entities. Interestingly, in patients with eosinophilic gastritis or duodenitis, a phase 2 trial with the anti-Siglec-8 antibody AK002 (lirentelimab) reduced gastrointestinal eosinophils and symptoms in these patients, although no continuation of the study has been released [131].

## 4. Concluding Remarks

The data described in this review highlight the huge potential of eosinophils as key neuro-immune players in the gastrointestinal tract but also expose the need to increase our understanding of gut eosinophil–neuronal crosstalk with still so many unexplored aspects of the influence of eosinophil activity on the nervous system and its dual functional consequences. Eosinophil plasticity allows the expression of a variety of molecules; therefore, it is necessary to determine how micro-environmental conditions modulate the expression of receptors to better understand their predisposition to respond to neural stimuli and the consequences of these interactions for gut homeostasis and disease. Eosinophil abilities to communicate and respond to nerves and other immune cells are indeed of great relevance, particularly in diseases where unknown mechanisms maintain a low state of inflammation together with a relevant degree of immune activation, as in DBGIs. Current available therapies against eosinophil-mediated inflammatory conditions are directed towards chemotactic prevention or survival impairment. In the coming years, we expect to see increasing functional and micro-anatomical evidence of the involvement of eosinophil-driven neuro-immune interactions that will undoubtedly contribute to the enrichment of the therapeutic landscape and the amelioration of gastrointestinal pathology as well as the promotion of gut health.

## Figures and Tables

**Figure 1 cells-11-01644-f001:**
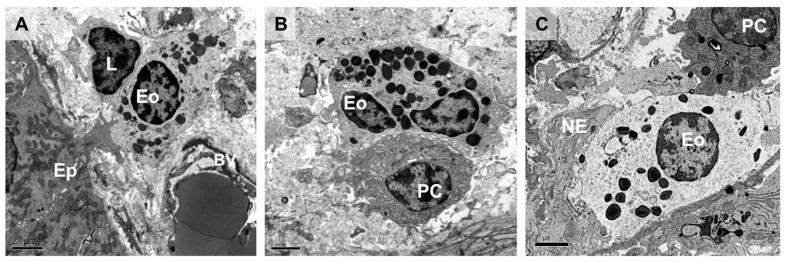
Intestinal eosinophils in close proximity to mucosal resident cells. Transmission electron micrographs of healthy human intestinal mucosa showing interactions of eosinophils with epithelial cells, nerve endings, and other immune cells. The eosinophil is identified based on morphological characteristics, mainly the cytoplasmic granules with well-defined electron-dense cores (crystalline cores) and a bilobed nucleus (not always observable under transmission electron microscopy). (**A**) Intestinal epithelial cell extending a protrusion to establish direct contact with a subepithelial eosinophil which is in contact with a lymphocyte. (**B**) Eosinophil physically interacts with a plasma cell in the intestinal mucosa, along the surface of the cell membrane. (**C**) Dual interaction of an eosinophil with a plasma cell and a free nerve ending along its cell membrane. Ep, epithelium; L, lymphocyte; Eo, eosinophil; BV, blood vessel; PC, plasma cell; NE, nerve ending. Bar: 2 µm.

**Figure 2 cells-11-01644-f002:**
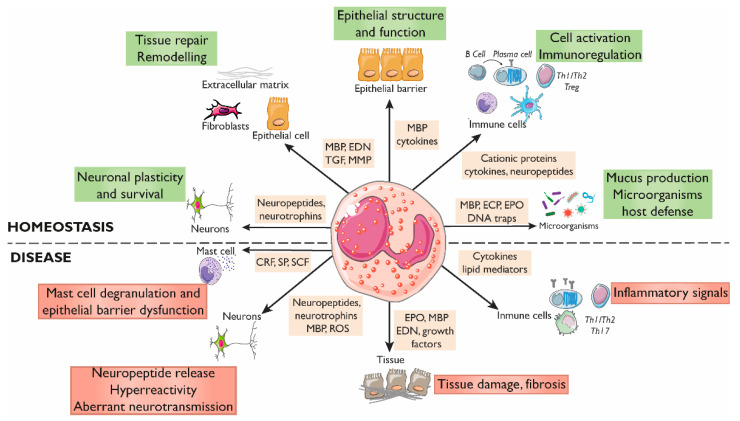
Roles of eosinophils in gastrointestinal homeostasis and disease. The interactions between eosinophils and other constituents of the intestinal mucosa (nerves, immune cells, endothelium, stroma, and luminal factors) mediate barrier integrity maintenance or tissue damage through the release of granule mediators. Major basic protein and eosinophil-derived cytokines contribute to barrier function through prostaglandin and ion secretion and mucus production. Eosinophils also impact tissue immuno-regulation and inflammation through cytokines and growth factors that promote adaptive and innate immunity, including a direct effect on mast cells in inflammatory or under stress conditions. Eosinophils also prime B cells for antigen-specific IgM production and sustain long-lived plasma cells. They participate in protection against infections through the release of DNA traps and the stimulation of mucus production through cationic proteins. In addition, eosinophils are also a source of a varied range of proteins and cytokines involved in fibrogenesis and angiogenesis that promote tissue remodeling, repair, and fibrosis. Eosinophils can regulate the function, survival, and development of nerve cells through the release of eosinophil-derived neuropeptides and neurotrophines, MBP, and ROS, and can promote hyperreactivity and aberrant neurotransmission in pathological conditions. CRF, Corticotropin releasing factor; ECP, Eosinophil cationic protein; EDN, Eosinophil-derived neurotoxin; EPO, Eosinophil peroxidase; MBP, Major basic protein; MMP, Matrix metalloprotease; ROS, Reactive oxygen species; SCF, Stem cell factor; SP, Substance P; TGF, Transforming growth factor.

**Figure 3 cells-11-01644-f003:**
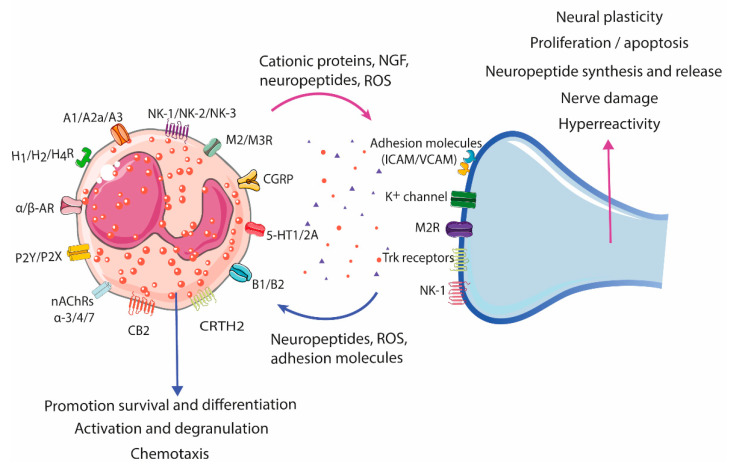
Eosinophil–neuron interaction. Eosinophils and neurons communicate bidirectionally, modulating functions in both cell types. Nerve cells can recruit eosinophils through the release of neuropeptides, cytokines, and chemokines, and activate degranulation through nerve contact facilitated by adhesion molecules (intercellular and neural cell adhesion molecules (ICAMs and NCAMs). The release of bioactive mediators by eosinophils, such as cationic proteins, ROS, and neuropeptides/neurotrophines, results in a variety of neuronal effects, including nerve growth, nerve damage, neuropeptide synthesis and release, and activation and sensitization, leading to hyperreactivity and aberrant neuropeptide release in pathological conditions. A1/A2a/A3, Adenosine receptor; AR, Adrenergic receptor; B1/B2, Bradykinin receptor; CB2, Cannabinoid receptor 2; CGRP, Calcitonin gene-related peptide; CRTH2, Chemoattractant receptor-homologous molecule expressed on Th2 cells; H1/H2/H4R, Histamine receptor H1, H2, H4; ICAM, Intercellular cell adhesion molecule; M2/M3R, Muscarinic receptor 2, 3; nAChRs α-3/-α4/α-7, Nicotinic acetylcholine receptor α-3, -α4, and α-7; NK-1/NK-2/NK-3, Tachykinin receptor 1, 2, 3; P2Y/P2X, Purinergic receptor 2Y, X; ROS, Reactive oxygen species; TrK, Tropomyosin receptor kinase; VCAM, Vascular cell adhesion molecule; 5-HT1/2A, 5-hydroxytryptamine 1, 2A.

**Table 1 cells-11-01644-t001:** Eosinophil mediators (modified from [23,24,25]).

Type of Molecule	Class	Function	Mediator
Cationic Proteins	Preformed	Host Defense/Barrier Function Homeostasis	ECP, EDN, EPO, MBP
Cytokines and growth factors	Preformed and *de novo* synthesis	Adaptive immunity	Th1: IFN-γ, IL-2, IL-12 Th2: IL-4, IL-5, IL-13, IL-9, IL-25 Th17: IL-17A, IL-17F Treg: IL-10, TGF-β
		Innate immunity	GM-CSF, IL-3, IL-4, IL-5, IL-13, SCF
		B cell class-switch and plasma cell maintenance	APRIL, IL-4, IL-6
		Tissue remodeling and repair	TGF-α, TGF-β, IL-1β, IL-13, PDGF-B, VEGF
Chemokines	Preformed and *de novo* synthesis	Recruitment of innate and adaptive immune cells	CCL3, CCL5, CCL6, CCL7, CCL8, CCL11, CCL13, CCL17, CCL22, CXCL9, CXCL10, IL-8
Lipid mediators	*De novo* synthesis	Pro-inflammatory	LTC4, PAF, PGE, PGF1, TxA2
		Resolution inflammation	PD1, RvE3
Neuropeptides and neurotrophines	Preformed and *de novo* synthesis	Nerve function, survival, and development	CGRP, CRF, NGF, NT-3, SP, VIP

APRIL, A proliferation-inducing ligand; CGRP, Calcitonin gene-related peptide; GM-CSF, Granulocyte macrophage colony-stimulating factor; CRF, Corticotropin releasing factor; ECP, Eosinophil cationic protein; EDN, Eosinophil derived neurotoxin; EPO, Eosinophil peroxidase; IL, Interleukin; LTC4, Leukotriene C4; MBP, Major basic protein; NGF, Nerve growth factor; NT-3, Neurotrophin-3; PAF, Platelet-activating factor; PD1, Protectin D1; PDGF-B, Platelet-derived growth factor subunit B; PGE, Prostaglandin E; PGF1, Prostaglandin F1; RvE3, Resolvin E3; SCF, Stem cell factor; SP, Substance P; TGF, Transforming growth factor; TxA2, Thromboxane A2; VEGF, Vascular endothelial growth factor; VIP, Vasoactive intestinal peptide.

## Data Availability

Not aplicable.

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
