# Peer review of "Eosinophils in the Gastrointestinal Tract: Key Contributors to Neuro-Immune Crosstalk and Potential Implications in Disorders of Brain-Gut Interaction"

_cells, 2022, doi:10.3390/cells11101644_

Round 1
Reviewer 1 Report
In this review article Salvo-Romero and colleagues discuss the role of eosinophils in intestinal homeostasis. Although I appreciate that the authors put in a lot of effort in writing the manuscript and designing the figures, I would propose to revise the manuscript based on the following suggestions before it can be considered for publication.
Major concerns:
The major content is only summarized in tables without sufficient discussion. Some of the content of the tables is debatable:
Table 1: Do we really believe Eosinophils express IL-5, IL-13, IL-9 and IL-25 (similar for Th1 and Th17 or cytokines) or VIP?
Table 2: The same is true for adrenergic receptors and VIP receptors. Do Eosinophils express so many adrenergic receptors? Which VIP receptor is expressed?
I would propose the authors reanalyze published RNA-sequencing data deposited in a publicly available database, for example, Immgen and compare the results to the content of the table.
In addition., the authors should critically discuss the findings presented in the tables and their limitations rather than list them without further comment.
Figure 3: It remains unclear which factor Eosinophils and neurons use for communication.
Minor concerns:
Some statements made are too general or unspecific:
L.77 …showing several cell types…
L.92 …under the control of specific transcription factors…
L.184 …upon different stimuli…
In vivo and in vitro is usually written in italic (L.146/147)?
Do secretomotor neurons release Ach in mice (L.258)?
Reviewer 2 Report
Re: Manuscript ID: cells-1687055
I congratulate the author on his well written review dealing with the intriguing role of eosinophils in the gastrointestinal tract. The different sections are well balanced and explored. Some minor changes are suggested to improve the paper.
Points of criticism
Line 108. Delete the comma.
Line 129. “de novo” in italics.
Line 146. “in vivo” in italics throughout the text.
Line 147. “in vitro” in italics throughout the text.
Line 158. Replace “(figure1)” with “(figure 1)”.
Figure 2. Replace “inmunoregulation” with “immunoregulation” and “inflamatory” with “inflammatory”.
Line 223. Replace “includes” with “include”.
Line 271. The abbreviation “GI” is only in this line.
Table 2. Replace “pro-or” with “pro- or”.
Line 304. Replace “Interleukine” with “Interleukin”.
Line 332. Replace “celllular” with “cellular”.
Line 344. Replace “stimulus” with “stimuli”.
Line 452. Replace “mechanisms this” with “mechanisms in this”.
Line 464. Replace “substance P” with “SP”.
Reviewer 3 Report
In the proposed paper, Salvo-Romero et al nicely reviewed the contribution of eosinophils in the gastrointestinal tract. The paper is well written, well illustrated and interesting but some points need to be added or modified.
Page 2 (Introduction):
The title of figure 1 should be changed. For the reviewer, the term “physically” is not adapted, it is more a proximity.
Lines 57-59: A more recent publication (O’Shea KM, Aceves SS, Dellon ES, Gupta SK, Spergel JM, Furuta GT, et al. Pathophysiology of Eosinophilic Esophagitis. Gastroenterology 2018;154(2):333‑4) that could be added.
Line 62: The authors cited the pathologies EGID, IBD and DBGI but did not cite Eosinophilic esophagitis (EoE). In all the manuscript, the place of EoE should be increased.
Page 3 (eosinophils in the gastrointestinal tract)
Line 95: The authors wrote “ with the only exception of the healthy esophagus”. Thus, it is important to introduce EoE in the first part of the paper.
Page 4 (eosinophils regulate the intestinal barrier function)
Lines 152-154 : The authors discussed about the ratio eosinophils/epithelial cells and mentioned that a low ratio is associated with a barrier disruption. Again, the exception of esophagus could be mentioned since the ratio is low in healthy esophagus. Are cut-off available for this ratio according to the part of gastrointestinal tract ?
Page 6 :
Line 207: IL-13 seems also to play a key role both in fibrosis rmodeling process and in the regulation of barrier function (Blanchard C, Mingler MK, McBride M, Putnam PE, Collins MH, Chang G, et al. Periostin facilitates eosinophil tissue infiltration in allergic lung and esophageal responses. Mucosal Immunol. juill 2008;1(4):289‑96 ; Sherrill JD, Kc K, Wu D, Djukic Z, Caldwell JM, Stucke EM, et al. Desmoglein-1 regulates esophageal epithelial barrier function and immune responses in eosinophilic esophagitis. Mucosal Immunol. mai 2014;7(3):718‑29.)
Page 6 (eosinophils modulate intestinal immune responses)
Line 248: Indeed, mast cells regulate the survival and activation of eosinophils but it could of interest to mention that mast cells are not necessarily required. In mast cell-deficient mouse, the number of eosinophils remains unaffected in a model of EoE (Niranjan R, Mavi P, Rayapudi M, Dynda S, Mishra A. Pathogenic role of mast cells in experimental eosinophilic esophagitis. American Journal of Physiology-Gastrointestinal and Liver Physiology. 15 juin 2013;304(12):G1087‑94).
Page 11 (Emerging role of eosinophils in disorders of brain-gut interaction)
Lines 419-423: an epithelial barrier dysfunction has also been described in EoE.
Page 20 :
Line 841: delete 114 that is duplicate
Round 2
Reviewer 1 Report
Major concerns:
Point 1: Which are the original populations demonstrating expression of the cytokines IL-5, IL-13, IL-9, and IL-25 (similar for Th1 and Th17 or cytokines)? I see only review articles cited here. In addition, if I check for example, the Immgen database, the expression of these cytokines is very low or absent in eosinophils. Therefore, I would recommend reanalyzing deposited RNA-sequencing data.
Point 2: CRTH2 is not the first gene that comes to my mind when I think about VIP receptors. Please at least discuss that.
Point 3:A critical discussion is mandatory since the expression of some of the cytokines and receptors is not reflected by the database (see the response to Point 1).
Minor concerns:
Point 3: If you believe Ach neurons are not well-defined in mice despite the many high-impact publications, please at least discuss the controversy.
